# Evaluation of Fermentability of Whole Soybeans and Soybean Oligosaccharides by a Canine In Vitro Fermentation Model

Hee Seong Kim [1], Evan C. Titgemeyer [2] and Charles Gregory Aldrich [1,*]

1 Grain Science and Industry Department, Kansas State University, Manhattan, KS 66506, USA; heeseong@ksu.edu
2 Animal Science and Industry Department, Kansas State University, Manhattan, KS 66506, USA; etitgeme@ksu.edu
* Correspondence: aldrich4@ksu.edu

**Abstract:** Soybean oligosaccharides (OS) have been recognized as a prebiotic that can be fermented in the colon, resulting in short-chain fatty acid (SCFA) production that can be used as an energy source for colonocytes, supporting cell differentiation and gut health. The objective was to determine the effects of WSBOS on in vitro fermentation, using dog feces as inoculum. Treatments included total dietary fiber (TDF) residues from WSB, soybean hulls (SH), pea fiber (PF), and beet pulp (BP), as well as WSB TDF residue plus soybean OS (WSBOS) and WSB TDF residue plus raffinose, stachyose, and verbascose (WSBRSV). Fresh fecal samples were collected from dogs and maintained in anaerobic conditions until substrate inoculation. Test tubes containing fiber sources and inoculum were incubated for 4, 8, and 12 h at 39 °C. Organic matter disappearance (OMD), pH, and SCFA were measured. The WSBOS and WSBRSV had greater ($p < 0.05$) OMDs than BP. Butyrate production was greatest ($p < 0.05$) for WSBOS (294.7 μmol/g) and WSBRSV (266.1 μmol/g), followed by BP (130.3 μmol/g) and WSB (109.2 μmol/g), and lowest ($p < 0.05$) for PF (44.1 μmol/g). The production of total SCFA was greatest ($p < 0.05$) for BP and WSBOS, followed by WSB, and lowest ($p < 0.05$) for PF. In conclusion, WSB has the potential as a prebiotic demonstrating greater butyrate production than BP in a canine in vitro fermentation model due to the fermentation of both OS and fiber in WSB. Further animal feeding studies are needed to determine the appropriate amount of WSB in canine diets.

**Keywords:** beet pulp; butyrate; dog nutrition; dietary fiber; short chain fatty acids; legume; prebiotics

## 1. Introduction

The pet food market continues to shift toward more premiumization and use of more whole ingredients with nutrition–health-related messages. Specifically, whole soybeans (WSB) can be a nutritious ingredient for dogs because they contain 38.5% crude protein and 20.9% crude fat (DM basis) [1]. However, WSB consist of approximately 8% soybean hulls [2], which are mostly fiber (63.8 to 81.2% total dietary fiber (TDF) [3]. In addition, soybeans contain oligosaccharides (OS) in significant quantities [4], which include galactosyl-sucrose OS such as raffinose, stachyose, and verbascose [5].

Prebiotic was defined as 'a nondigestible food ingredient that beneficially affects the host by selectively stimulating the growth and/or activity of one or a limited number of bacteria in the colon, and thus improves host health' in 1995 [6,7]. Galactooligosaccharides were one of the established prebiotics along with galactan, fructooligosaccharides, fructans, lactulose, oligofructose, and inulin [6,8]. The soybean OS have been recognized as prebiotics because they promote the growth of beneficial bacteria in the colon, mainly *Bifidobacterium* spp. [9].

Indigestible OS and soluble non-starch polysaccharides (NSP) in soybeans have been indicated as anti-nutritional factors in animal diets that may negatively affect diet digestibility [4,10]. There have been numerous reports that described the relationship between the

poor growth performance in monogastric animals (broilers and weaning pigs) when they were fed diets containing soybean meal that had a high fiber content, including OS and soluble NSP [11]. Moreover, due to fermentation in the large intestine, the production of gas, lactate, and short-chain fatty acids (SCFA) may result in softer feces and flatulence in dogs [12]. However, these indigestible compounds may be beneficial in the gastrointestinal tract in dogs if they are provided in the diet at an ideal dose.

Soluble fiber is known to be degraded by microbiota in the colon, resulting in SCFA production [13]; however, differences in fiber composition have a big impact on fermentation level and fermentation end-product profiles. Beet pulp (BP) is considered a standard fiber source in pet foods, with its soluble fibers providing benefit to colonic fermentation in dogs [14,15]. Pea fiber (PF) has been evaluated in an in vitro model with canine fecal inoculum, and it was intermediate in the production of SCFA compared to beet pulp and cellulose [16]. Soybean hulls (SH) were evaluated in an in vitro model with canine fecal inoculum [17], and they were intermediate in their production of total SCFA, which was lower than fructans but greater than potato starch. Prior to conducting an animal feeding trial, it is important to evaluate the fermentability of soybean OS using an in vitro model [15,16,18]. The objectives of this study were to determine the fermentation characteristics of WSB fiber residues (dietary fiber, OS, and SH) compared to traditional fiber sources, such as BP and PF, using an in vitro model with dog fecal inoculum to gain preliminary information about how WSB might function in canine diets. We hypothesized that a mixture of soybean fiber and soybean OS in ratios found in WSB would have fermentability similar to BP but greater than PF.

## 2. Materials and Methods

The dog feeding experiment was approved by the Kansas State University Institutional Animal Care and Use Committee (IACUC) under protocol #4566.

### 2.1. Fiber Sources and Treatment Preparation

The procedures for the preparation of fiber sources were an adaptation of published methods [16]. The BP (Fairview Mills, Seneca, KS, USA), PF, and SH (Lorschters, Bern, KS, USA) fiber sources were selected to be compared with WSB residues because they have been evaluated in previous companion animal feeding studies (PF, [19]; BP, [20]). The SH and PF were ground in a laboratory-fixed blade impact mill (Retsch ZM200, Haan, Germany) to pass a 0.5 mm screen. The WSB was purchased from a local grain elevator (MKC, Manhattan, KS, USA) and ground with a hammer mill (model 18-7-300; Schutte Buffalo, NY, USA) to pass through a 1.19 mm screen. The soybean OS (WSBOS) were provided by a regional soybean processing company (Prairie AquaTech, Brookings, SD, USA). Individual galactosyl-sucrose OS (stachyose, raffinose, and verbascose) were purchased from chemical supply companies (Verbascose > 95%, Neogen, Lansing, MI, USA; Raffinose > 98%, Tokyo Chemical Industry, Tokyo, Japan; Stachyose hydrate from stachys tuberifera $\geq$ 90%, Chem-Impex International, Wood Dale, IL, USA).

Prior to the incubation, fiber residues from PF, BP, SH, and WSB samples were isolated using a total dietary fiber assay kit (Neogen, catalog no. K-TDFR-200A) to simulate the digestion in the small intestine of the dogs. As WSB contains significant quantities of fat, the finely ground WSB were defatted before the isolation of total dietary fiber (TDF). In this step, 100 g of ground WSB samples were placed in a 1 L beaker with a stir rod on a stirring plate under the exhaust hood. To this, 400 mL of hexane was added into the beaker and stirred for 20 min. Using a porcelain Buchner funnel and a Whatman (Marlborough, MA, USA) ashless filter paper (grade 541), the hexane mixture was filtered, and residues were dried at room temperature overnight under the hood to evaporate the hexane. The dried residues were used for TDF isolation. The isolation process followed the total dietary fiber assay protocol (Neogen, catalog no. K-TDFR-200A) with minor modifications. The PF, BP, SH, and defatted WSB were digested with α-amylase, protease, and amyloglucosidase in bulk. A 10 g sample was mixed with 400 mL MES-TRIS buffer solution and 500 µL

heat-stable $\alpha$-amylase and then placed in a shaking (75 rpm) water bath at 98–100 °C for 45 min. The sample beakers were cooled to 60 °C and mixed with 1 mL protease and placed in a shaking (75 rpm) water bath set at 60 °C for 45 min. The samples were then removed from the water bath and mixed with 50 mL of 0.561 N HCl and placed on a stirring plate. The pH was adjusted to 4.5–4.7, adding either additional 5% NaOH or 5% HCl. To this, 2 mL of amyloglucosidase solution was added while stirring, and the samples were incubated in a shaking (75 rpm) water bath at 60 °C for 45 min. Then, 2250 mL of the pre-heated (60 °C) 95% ethanol was added and precipitated overnight. Using a porcelain Buchner filter and a square piece of fabric (pore size 100 microns), the precipitated solution was filtered and sequentially washed with 78% (*vol/vol*) ethanol, 95% (*vol/vol*) ethanol, and acetone. The fabric containing the residue was dried overnight in a convection oven at 105 °C. The dried residue was referred to as TDF residue from each fiber source and was used as a treatment sample (SH, PF, BP, and WSB) for the in vitro fermentation study.

The WSBOS treatment samples were prepared by mixing WSB TDF residues with soybean OS (a commercial product from Prairie AquaTech), based on the ratio of the actual TDF and total OS content of the raw WSB (TDF:OS on DM basis = 19.8:8.02) (Table 1). The WSBOS treatment represented the case for a dog fed WSB because WSBOS contains both the TDF residue and the OS that were filtered and lost during fiber isolation from WSB. The WSBRSV treatment samples were prepared by mixing WSB TDF residues with each oligosaccharide (raffinose, stachyose, and verbascose) in a corresponding portion of the oligosaccharides from analytical results obtained from the raw WSB (TDF/raffinose/stachyose/verbascose on a DM basis = 19.8:0.56:3.16:0.01) (Table 1). The difference between WSBOS and WSBRSV would largely be sucrose. The blank treatment did not contain any substrate and was utilized to assess fermentation from substrate in the inoculum.

**Table 1.** Oligosaccharide concentrations in whole soybeans.

| Oligosaccharide | % of Dry Matter |
| --- | --- |
| Sucrose | 4.28 |
| Raffinose | 0.56 |
| Stachyose | 3.16 |
| Verbascose | 0.01 |

*2.2. Dog Donors and Inoculum Preparation*

The procedures for the preparation of the inoculum and the incubation of the fibrous substrates were an adaptation of published methods [16]. The Beagle dog donors (body weight; 7.9 ± 1.38 kg) were individually housed in the Large Animal Research Center of Kansas State University (Manhattan, KS, USA). The commercial diet (Table 2) was provided twice daily for each dog to maintain their body weight. Feces for the preparation of the inoculum were collected fresh within 15 min after defecation. Feces from the 3 dogs were collected immediately into plastic bags, and the air was removed from the bag to avoid exposure to aerobic conditions. The bags were placed in an insulated container that contained warm water (37 °C) to maintain temperature during transport to the lab. Samples of 33 ± 1 g of each feces was pooled together, making the total of 100 g combined fecal samples. The pooled fresh fecal samples were diluted 1:10 in an anaerobic dilution solution (Table 3) and purged with $CO_2$. The solution was blended well until most of the fecal lumps were dispersed. The solution was then filtered through 4 layers of cheese cloth under purging $CO_2$. The filtered solution was used as the inoculum for the fermentations.

**Table 2.** Analyzed nutrient composition of the commercial diet [1] fed to dogs during and before collection of fresh feces.

| Nutrient | Diet |
|---|---|
| Dry matter, % | 91.4 |
| Ash, % of dry matter | 8.0 |
| Crude protein, % of dry matter | 31.2 |
| Acid hydrolyzed ether extract, % of dry matter | 15.2 |
| Total dietary fiber, % of dry matter | 14.0 |

[1] Ingredients: ground corn, chicken meal, corn gluten meal, rice flour, porcine meat and bone meal, dried plain beet pulp, poultry fat preserved with BHA, porcine animal fat preserved with BHA and citric acid, brewers dried yeast, hydrolyzed poultry by-products aggregate, spray dried animal blood cells, dried egg product, dried whey, L-lysine, salt, dicalcium phosphate, soybean oil, natural flavor, potassium chloride, calcium carbonate, choline chloride, pyridoxine hydrochloride, DL-methionine, menadione dimethyl pyrimidinol bisulfite (source of vitamin K), cholecalciferol (form of vitamin D3), lecithin, biotin, vitamin A acetate, DL-alpha tocopheryl acetate (form of vitamin E), ferrous sulfate, inositol, preserved with mixed tocopherols, zinc oxide, calcium pantothenate, folic acid, thiamine mononitrate, calcium iodate, ethoxyquin (a preservative), riboflavin supplement, nicotinic acid, manganous oxide, vitamin B-12 supplement, copper sulfate, cobalt carbonate.

**Table 3.** Composition of inoculation medium and anaerobic dilution solutions.

| Solution | Medium | Anerobic Dilution |
|---|---|---|
| Solution A [1], mL | 330.0 | 37.50 |
| Solution B [2], mL | 330.0 | 37.50 |
| Mineral solution [3], mL | 10.0 | - |
| Vitamin solution [4], mL | 10.0 | - |
| Folate-biotin solution [5], mL | 5.0 | - |
| Riboflavin solution [6], mL | 5.0 | - |
| Hemin solution [7], mL | 2.5 | - |
| Resazurin solution [8], mL | 1.0 | 1.00 |
| Water, mL | 296.0 | 854.00 |
| Yeast extract, g | 0.5 | - |
| Trypticase, g | 0.5 | - |
| $Na_2CO_3$, g | 4.0 | 6.37 |
| Cysteine hydrochloride, g | 0.5 | 0.50 |

[1] Solution A—5.4 g sodium chloride, 5.4 g ammonium sulfate, 2.7 g potassium phosphate monobasic anhydrous, 0.18 g calcium chloride dihydrate, 0.12 g magnesium chloride hexahydrate, 0.06 g manganese chloride tetrahydrate, 0.06 g cobalt chloride hexahydrate, to 1 L with distilled water. [2] Solution B—2.7 g potassium phosphate dibasic anhydrous to 1 L with distilled water. [3] Mineral solution—500 mg of ethylenediaminetetraacetic acid, 200 mg iron (II) sulfate heptahydrate, 30 mg m-phosphoric acid, 20 mg cobalt chloride hexahydrate, 10 mg zinc sulfate heptahydrate, 3 mg manganese chloride tetrahydrate, 3 mg sodium molybdate dihydrate, 2 mg nickel (II) chloride hexahydrate, 1 mg copper (II) chloride dihydrate, to 1 L with distilled water. [4] Vitamin solution—Weigh 100 mg thiamin hydrochloride, 100 mg pantothenic acid, 100 mg niacin, 100 mg pyridoxine hydrochloride, 10 mg ammonium carbonate, 5 mg 4-aminobenzoic acid, 0.25 mg vitamin B-12, to 1 L with distilled water. Added to the medium by filter sterilization after other reagents were sterilized in autoclave. [5] Folate-biotin solution—100 mg ammonium carbonate, 10 mg folic acid, 2 mg biotin, to 1 L with distilled water. [6] Riboflavin solution—130 mg HEPES, 1 mg riboflavin, to 1 L with distilled water. [7] Hemin solution—50 mg hemin, 40 mg sodium hydroxide, to 100 mL with distilled water. [8] Resazurin solution—100 mg resazurin to 100 mL with distilled water.

*2.3. Canine In Vitro Microbial Fermentation*

The treatment samples were subjected to in vitro microbial fermentation, as described by [16], with some modifications. Briefly, $0.3 \pm 0.0001$ g of the treatment samples were weighed in triplicate in 50 mL conical centrifuge tubes for each one of the 3 time points (4, 8, 12 h). In addition to the tubes with the fiber samples, tubes without any fiber samples for each time point in triplicate were used as blanks. To each tube, 26 mL of media solution was added. Next, each tube was flushed with $CO_2$ and closed with a rubber stopper equipped with a 1-way Bunsen valve. Tubes were then placed in the refrigerator overnight to allow substrates to hydrate. On the following day, the samples were placed in a shaking (60 rpm) water bath at 39 °C for 45 min prior to inoculation.

Tubes were inoculated with 4 mL of the prepared inoculum using a repeater pipette, starting with tubes from time 12, 8, and, lastly, 4 h. After inoculation, tubes were flushed

with $CO_2$, closed with a rubber stopper equipped with a 1-way Bunsen valve, and incubated in a shaking (30 rpm) water bath at 39 °C for the predetermined time points. The tubes were vortexed every 2 h. At each incubation timepoint, a 1 mL subsample from each tube was transferred to a 2 mL microcentrifuge tubes and mixed with 0.25 mL of 25% (*wt/vol*) m-phosphoric acid for deproteinization. The microcentrifuge tubes were frozen at −20 °C until SCFA analysis. The pH of the remaining solutions in the 50 mL conical centrifuge tubes were measured by inserting a calibrated glass-electrode pH probe (FC240B, Hanna Instruments, Smithfield, RI, USA) directly into the sample. All the remaining solution in the 50 mL conical centrifuge tube were transferred to a 400 mL beaker, mixed with 112 mL of 95% ethanol, and allowed to precipitate overnight at room temperature.

On the following day, the solutions were filtered using pre-weighed dried Whatman (Marlborough, MA, USA) ashless filter paper (grade 541) and a porcelain Buchner funnel with a vacuum pump. The filtered residues were rinsed twice with 10 mL of 95% ethanol and twice with 10 mL of acetone. Next, the filter paper containing the residues were put in a 50 mL beaker and dried in a convection oven overnight at 105 °C. The dry weights of the filter and residue were recorded the following day. After samples were weighed for DM, the 50 mL beakers containing the filter and residue were placed in the muffle furnace at 450 °C overnight and weighed on the next day for ash corrections.

Nutrient compositions were variable across the fiber sources and the TDF residues that were used as substrates in the study (Table 4). The TDF content of the PF was the highest (72.9%), followed by SH (67.9%), BP (61.1%), and WSB (21.5%) on a dry matter basis. The IDF content of the PF was the highest (68.0%), following by SH (58.3%), BP (36.5%), and WSB (19.4%) on a dry matter basis. The WSB contained more CP (38.5%) on a DM basis than the other fiber sources, such as SH (17.0%), BP (15.2%), and PF (14.0%). The CP content of defatted WSB TDF residue was the highest (38.0%) and that of PF was the lowest (8.4%).

**Table 4.** Nutritional composition of fiber sources (WSB, whole soybean; DWSB, defatted whole soybean; SH, soy hull; BP, beet pulp; PF, pea fiber) and the total dietary fiber (TDF) residues isolated from the fiber sources and used as substrates.

| Item | WSB | DWSB | SH | BP | PF |
|---|---|---|---|---|---|
| Fiber sources | | | | | |
| Dry matter, % | 92.5 | 91.9 | 98.0 | 91.9 | 92.4 |
| | | -dry matter basis- | | | |
| Organic matter, % | 94.9 | 93.7 | 95.1 | 94.6 | 96.9 |
| Crude protein, % | 38.5 | 47.8 | 17.0 | 15.2 | 14.0 |
| Total dietary fiber, % | 21.5 * | 25.8 | 67.9 | 61.1 | 72.9 |
| Insoluble dietary fiber, % | 19.4 * | 23.3 | 58.3 | 36.5 | 68.0 |
| Soluble dietary fiber [1], % | 2.1 * | 2.5 | 9.6 | 24.6 | 4.9 |
| TDF residues | | | | | |
| Dry matter, % | n.d. [2] | 90.5 | 90.9 | 89.3 | 89.7 |
| | | -dry matter basis- | | | |
| Organic matter, % | n.d. | 94.9 | 97.0 | 94.0 | 97.7 |
| Crude protein, % | n.d. | 38.0 | 12.6 | 12.9 | 8.4 |

[1] Calculated as total dietary fiber—insoluble fiber. [2] n.d. means not determined. * Calculated values using analysis results for defatted WSB samples.

### 2.4. Determination of Organic Matter Disappearance (OMD) and Chemical Analysis

The OMD (%) were calculated as follows:

$$\text{OMD}(\%) = \left(1 - \frac{\text{organic matter residue(g)} - \text{organic matter blank(g)}}{\text{initial organic matter(g)}}\right) \times 100\% \quad (1)$$

where in organic matter (OM) residue is OM in the sample after the incubation and filtration in g, OM blank is OM in the blank after incubation and filtration in g, and Initial OM is the initial OM in the sample prior to incubation in g.

All chemical analyses were performed in duplicate unless otherwise specified. The fiber samples were ground using a fixed blade laboratory mill (Retch, type ZM200, Haan, Germany), fitted with a 1.0 mm screen, and stored in lidded glass jars in preparation for chemical analysis. The ground WSB, defatted WSB, SH, BP, PF, and TDF residues were analyzed for DM, OM, and ash (AOAC methods 934.01 and 942.05). Crude protein (CP) content of the samples was determined by the Dumas combustion method (AOAC 990.03), using a nitrogen analyzer (FP928, LECO Corporation, Saint Joseph, MI, USA). The insoluble dietary fiber (IDF) and TDF content of the samples were measured following the standard procedure from the Total Dietary Fiber Assay Kit (K-TDFA-200A, Neogen). The soluble dietary fiber (SDF) was calculated subtracting IDF concentration from the TDF concentration. The WSB sample was analyzed by the Agricultural Experiment Station Chemical Laboratories (Columbia, MO, USA) for oligosaccharides (sucrose, stachyose, raffinose, and verbascose), as described by [21].

For SCFA analysis, samples were thawed and centrifuged at 20,000 g for 15 min. The supernatant was collected and filtered through a 0.2 μm PTFE syringe filter. The SCFA contents from the filtered samples were analyzed by gas–liquid chromatography [22], using a capillary column (BP-FATWAX UI, Agilent G3903-63008, 30 m × 0.25 mm × 0.25 μm; Agilent Technologies, Santa Clara, CA, USA). The system was equipped using helium as a carrier gas, with a flow rate of 40 cm/s, utilizing a 15:1 split ratio injector, with an injection size of 0.5 μL. A flame ionization detector was configured with hydrogen as the makeup gas with a flow rate of 25 mL/min. The detector and inlet temperatures were set at 250 °C, and the initial oven temperature was set to 80 °C with a ramp rate of 10 °C/min to 200 °C for a total run time of 15 min. The peak area of chromatograms was determined using integrative software (GC solution version 2.42.00, Shimadzu, Kyoto, Japan). The concentrations of SCFA in the supernatant of the fermented samples were quantified by comparing the sample peak area to 10 mm standards (Volatile Free Acid Mix, Sigma-Aldrich, St. Louis, MO, USA). Production of SCFA was calculated per unit of substrate DM. The average SCFA concentrations from the blank tubes were subtracted to correct any artifacts that the inoculum might have had.

### 2.5. Statistical Analysis

The experiment had 7 treatments (including the blank as a treatment) × 3 timepoints × 3 replicates. It was performed with a completely randomized design, with 50 mL conical centrifuge tubes as experimental units. The OMD, pH, and SCFA data were subjected to ANOVA for a completely randomized design with factorial arrangement of the substrate and time factors using the general linear model procedure of SAS (v. 9.4; SAS Inst. Inc., Cary, NC, USA). Differences of least square means were assessed using Tukey's post hoc test for multiple comparisons. Results were considered significant at $p < 0.05$.

## 3. Results

### 3.1. OMD and pH

For the OMD and pH, the 12 h time point, which is the maximum time for the fermentation in this trial, will be discussed unless otherwise specified (Table 5). The BP was regarded as the reference because it is a prominent fiber source used in commercial dog food, and its fermentability by dogs is well understood [16]. Compared to BP (41.0%), WSBOS (60.2%) had a higher OMD throughout all the time points ($p < 0.05$). The WSBRSV treatment (43.6%) had a less OMD than WSBOS (60.2%) throughout all the time points ($p < 0.05$). The WSB had a lower OMD ($p < 0.05$) than BP; however, WSB (37.6%) had a higher OMD than SH (20.4%) and PF (18.6%). The pH was affected by the different fiber sources in the current study. The BP and WSBRSV had the lowest pH among the treatments

($p < 0.05$), and the pH for the PF (7.05), WSB (6.90), SH (7.07), and WSBOS (7.01) did not differ and were lower than the blank pH (7.50).

**Table 5.** Least square means of organic matter disappearance (OMD, %) and pH of fermented fiber sources inoculated with dog feces for 4, 8, and 12 h (PF, pea fiber; WSB, whole soybeans; BP, beet pulp; SH, soy hulls; WSBRSV, WSB plus raffinose, stachyose, and verbascose; WSBOS, WSB plus soy oligosaccharides).

| Incubation Time, h | Blank | PF | BP | SH | WSB | WSBRSV | WSBOS | SEM [1] | *p*-Value |
|---|---|---|---|---|---|---|---|---|---|
| OMD, % | | | | | | | | | |
| 4 h | . | 15.8 [d] | 39.3 [b] | 19.6 [d] | 28.8 [c] | 37.2 [b] | 58.3 [a] | 1.70 | <0.0001 |
| 8 h | . | 19.3 [c] | 41.2 [b] | 18.5 [c] | 34.9 [b] | 38.6 [b] | 55.2 [a] | 1.34 | <0.0001 |
| 12 h | . | 18.6 [d] | 41.0 [b] | 20.5 [d] | 37.6 [c] | 43.6 [b] | 60.2 [a] | 0.60 | <0.0001 |
| pH | | | | | | | | | |
| 4 h | 7.05 [a] | 6.94 [ab] | 6.49 [d] | 6.82 [bc] | 6.79 [bc] | 6.64 [cd] | 6.71 [cd] | 0.046 | <0.0001 |
| 8 h | 7.05 [a] | 6.99 [a] | 6.60 [b] | 6.95 [ab] | 6.96 [ab] | 6.76 [ab] | 6.92 [ab] | 0.080 | 0.0193 |
| 12 h | 7.50 [a] | 7.05 [b] | 6.66 [d] | 7.07 [b] | 6.90 [bc] | 6.74 [cd] | 7.01 [b] | 0.041 | <0.0001 |

[1] SEM = standard error of the mean. [abcd] Means with different lowercase letters in the same row are significantly different at $p < 0.05$.

### 3.2. Short-Chain Fatty Acids

The descriptions for SCFA production presented here are based on the 12 h time point unless otherwise specified. Acetate production was greatest for BP, followed by WSBOS and WSBRSV, and lowest for PF and SH ($p < 0.05$; Table 6). Propionate production was greatest ($p < 0.05$) for BP, WSBRSV, and WSBOS, followed by WSB, PF, and SH. However, butyrate production was greatest ($p < 0.05$) for WSBOS (294.7 µmol/g) and WSBRSV (266.1 µmol/g), followed by BP (130.3 µmol/g) and WSB (109.2 µmol/g), and lowest ($p < 0.05$) for PF (44.1 µmol/g). The production of total SCFA was greatest ($p < 0.05$) for BP and WSBOS, followed by WSB, and lowest ($p < 0.05$) for PF.

**Table 6.** Least square means of short-chain fatty acid (SCFA) production from fermented fiber sources (PF, pea fiber; WSB, whole soybeans; BP, beet pulp; SH, soy hulls; WSBRSV, WSB plus raffinose, stachyose, and verbascose; WSBOS, WSB plus soy oligosaccharides) inoculated with dog feces for 4, 8, and 12 h, expressed as mmol/g of substrate dry matter.

| Fermentation Time, h | PF | BP | SH | WSB | WSBRSV | WSBOS | SEM [1] | *p*-Value |
|---|---|---|---|---|---|---|---|---|
| Acetate, µmol/g of substrate | | | | | | | | |
| 4 | 458 [d] | 1876 [a] | 794 [c] | 671 [cd] | 1232 [b] | 1411 [b] | 54.5 | <0.0001 |
| 8 | 616 [e] | 2172 [a] | 873 [de] | 1009 [d] | 1476 [c] | 1858 [b] | 60.4 | <0.0001 |
| 12 | 832 [e] | 2844 [a] | 1060 [de] | 1415 [cd] | 1817 [bc] | 2123 [b] | 86.2 | <0.0001 |
| Propionate, µmol/g of substrate | | | | | | | | |
| 4 | 176 [d] | 482 [bc] | 314 [cd] | 243 [d] | 570 [ab] | 698 [a] | 37.1 | <0.0001 |
| 8 | 228 [d] | 606 [b] | 326 [c] | 356 [c] | 657 [b] | 923 [a] | 18.1 | <0.0001 |
| 12 | 296 [b] | 923 [a] | 399 [b] | 468 [b] | 835 [a] | 992 [a] | 45.6 | <0.0001 |
| Butyrate, µmol/g of substrate | | | | | | | | |
| 4 | 27 [b] | 66 [b] | 64 [b] | 77 [b] | 205 [a] | 249 [a] | 15.2 | <0.0001 |
| 8 | 32 [e] | 78 [d] | 50 [e] | 105 [c] | 220 [b] | 308 [a] | 5.4 | <0.0001 |
| 12 | 44 [c] | 130 [b] | 63 [bc] | 109 [bc] | 266 [a] | 295 [a] | 16.6 | <0.0001 |

**Table 6.** *Cont.*

| Fermentation Time, h | PF | BP | SH | WSB | WSBRSV | WSBOS | SEM [1] | *p*-Value |
|---|---|---|---|---|---|---|---|---|
| Isobutyrate, µmol/g of substrate | | | | | | | | |
| 4 | 2.8 [c] | 4.0 [bc] | 11.9 [bc] | 10.7 [bc] | 15.4 [ab] | 25.0 [a] | 2.41 | 0.0003 |
| 8 | 4.4 [c] | 4.1 [c] | 8.0 [c] | 14.6 [b] | 18.0 [b] | 35.9 [a] | 1.38 | <0.0001 |
| 12 | 5.7 [c] | 12.7 [bc] | 11.5 [bc] | 14.2 [bc] | 19.0 [b] | 33.8 [a] | 2.31 | <0.0001 |
| Isovalerate, µmol/g of substrate | | | | | | | | |
| 4 | 1.9 [c] | 4.0 [c] | 14.3 [bc] | 13.0 [bc] | 20.8 [ab] | 31.5 [a] | 3.03 | 0.0002 |
| 8 | 2.9 [c] | 2.4 [c] | 8.5 [c] | 18.7 [b] | 23.0 [b] | 42.2 [a] | 1.65 | <0.0001 |
| 12 | 4.8 [d] | 10.3 [cd] | 12.2 [bcd] | 19.3 [bc] | 24.4 [b] | 40.5 [a] | 2.96 | <0.0001 |
| Valerate, µmol/g of substrate | | | | | | | | |
| 4 | 1.5 [c] | 3.4 [bc] | 3.5 [bc] | 2.5 [c] | 7.2 [ab] | 9.5 [a] | 0.93 | 0.0004 |
| 8 | 7.9 [c] | 7.6 [c] | 9.3 [c] | 13.3 [c] | 22.8 [b] | 46.6 [a] | 1.38 | <0.0001 |
| 12 | 14.3 [d] | 27.4 [cd] | 18.8 [d] | 45.7 [bc] | 63.3 [ab] | 79.0 [a] | 4.32 | <0.0001 |
| Total SCFA, µmol/g of substrate | | | | | | | | |
| 4 | 667 [c] | 2435 [a] | 1201 [b] | 1017 [bc] | 2050 [a] | 2424 [a] | 111.4 | <0.0001 |
| 8 | 891 [d] | 2869 [a] | 1274 [cd] | 1516 [c] | 2417 [b] | 3214 [a] | 81.4 | <0.0001 |
| 12 | 1196 [d] | 3948 [a] | 1565 [cd] | 2071 [c] | 3025 [b] | 3563 [ab] | 155.2 | <0.0001 |

[1] SEM = standard error of the mean. [abcde] Means with different lowercase letters in the same row are significantly different at $p < 0.05$.

## 4. Discussion

### 4.1. Nutritional Compositions

Sucrose, raffinose, and stachyose are major soluble sugars in soybean seeds [5]. Our WSB had similar TDF, sucrose, raffinose, and stachyose values to those previously reported by [23]. During the isolation of fiber from WSB, OS should not have been recovered in the TDF residue. Either soy OS or combinations of raffinose, stachyose, and verbascose were added to replace the loss of OS in the treatment preparations for WSBOS and WSBRSV. The TDF, IDF, and SDF contents of the PF, SH, and BP were also similar to the previous published data [3,16,20]. Dietary fiber and raffinose family oligosaccharides resisted hydrolysis by endogenous enzymes in the small intestine but may have been fermented by microbes in the colon to $CO_2$, $H_2$, ammonia, SCFA, and lactate. The source and solubility of the fiber determined the fermentation characteristics of the intestinal microbiota [24,25].

### 4.2. OMD and pH

Even though WSB contained less SDF than the SH and PF, the TDF residues of the defatted WSB contained higher CP, and this might have contributed to the higher OMD. The SH OMD reached its maximal level by 4 h. The major constituents of the total dietary fiber in SH were cellulose (30–50%), hemicellulose (15–25%), and pectin (6–15%) [26]. The high IDF content and the physical structure of SH likely contributed to impede anaerobic microorganisms to extend fermentation past 4 h. The fact that sucrose was present in WSBOS and not in WSBRSV might have partially contributed to the higher OMDs for the WSBOS than the WSBRSV. However, as the OS would not be recovered in the fermentation residues due to a small molecular size, even if they were not fermented, the OMDs of the WSBOS and WSBRSV were not necessarily reflective of the extent of fermentation.

The production of the SCFA via fermentation of carbohydrates by the gut bacteria reduces gastrointestinal luminal pH, which directly limits pathogen growth [27]. In an in vitro fermentation study using fecal samples from growing pigs [25], total gas production increased, and the pH of the fermentation substrate decreased as the SDF ratio increased compared to the IDF. Our results were similar to the research of [25] because the BP had the highest amount of SDF and the highest SDF to IDF ratio among the fibers, and this led to the lowest pH. However, the range of the pH difference was small. It is difficult to determine fermentability indirectly by pH measurement because substantial amounts

of buffer solutions were added in the in vitro fermentation procedure, and the substrates can provide buffering as well. We measured the SCFA concentrations as a more definitive measure of fermentation.

### 4.3. Short-Chain Fatty Acids

Some of the health benefits produced by dietary fibers are the production of fermentative end products and changes in the gastrointestinal microbiota [27]. The main bacterial fermentative end products are SCFA and the gases $H_2$ and $CO_2$; SCFA are an important indicator of fermentation in the colon [28]. The fermented end product profile depends on the substrate source and the microbial ecology in the colon [29]. According to previous work, pectin yields high acetate [30], gum yields high propionate [31], and resistant starch, lactose, and soybean oligosaccharides yield high butyrate concentrations after microbial fermentation with fecal inoculum [32]. The higher ratio of SDF to IDF in the dietary fibers increased in the concentrations of lactic acid, formic acid, and acetic acid, whereas the concentrations of propionic acid and butyric acid were greater in the low SDF ratio group in an in vitro fermentation experiment using pig fresh fecal inoculum [25]. The SCFA are used as an energy source for colonocytes and enterocytes and influence gastrointestinal epithelial cell integrity [28]. Acetate is absorbed and transported by the portal vein and used as a fuel for tissues throughout the body. Propionate is either taken up by the liver and converted to glucose [33] or locally utilized [34]. Butyrate is the major fuel source for colonocytes [29], increases colonocyte proliferation [35], and increases the mucin secretion in the large intestine [36]. Moreover, butyrate also influences various cellular functions affecting colonic health, such as anticarcinogenic and anti-inflammatory pathways, affects the intestinal barrier, and decreases oxidative stress [37]. Wächtershäuser and Stein (2000) [38] suggested that increasing luminal butyrate concentrations may be an appropriate means to ameliorate symptoms of inflammatory bowel diseases.

The soybean cell wall contains pectinic acid polysaccharides that contain uronic acids and neutral polysaccharides [39]. Yamaguchi et al. (1996) [40] found that pectic polysaccharides in soybeans had a similar molecular weight and galacturonan structure to that of fruit pectin. Galactose and arabinose were the main components in each of the polysaccharides. The WSB, WSBRSV, and WSBOS, which contained this pectinic acid group, had higher acetate and butyrate productions than the PF in this study. Swanson et al. (2001) [13] also found that citrus pectin produced higher amounts of acetate, butyrate, and total SCFA than pea hulls. The primary oligosaccharides found in the soybeans were galactooligosaccharides. Hernot et al. (2009) [30] reported that the galactooligosaccharides produced large quantities of SCFA, particularly butyrate, in an in vitro fermentation system. As we added soybean oligosaccharides to the WSB TDF residues, the butyrate and total SCFA production for WSBRSV and WSBOS was higher than for WSB. This was expected.

The colonic microflora might have degraded the NSP that remained in the WSB, SH TDF residues, and synthesized SCFA. Bakker et al. (1998) [41] found that the soybean hulls had more extensively fermented NSP than cellulose, yielding a higher amount of acetate, propionate, and butyrate in pigs. For our butyrate productions, WSBRSV had 144% the production of WSB, whereas WSBOS had 170% the production of WSB. These findings provided evidence that soybean galactooligosaccharides are fermented in the colon of dogs and yield a substantial amount of butyrate compared to acetate or propionate. Lan et al. (2007) [32] reported that stachyose and raffinose produced higher butyrate contents than soybean meal oligosaccharides when inoculated with the caecal contents from broilers for an in vitro fermentation model. The OS may have more potential than the soluble fibers in WSB to serve as substrate for butyrate production. The inclusion of WSB in diets will provide both WSB TDF and OS.

Hore and Messer (1968) [42] found that sucrase was present in the small intestine of dogs. However, sucrase levels are low in dogs throughout their lives [43,44]. Buddington et al. (2003) [45] reported that the activities of sucrase increased after birth in Beagle dogs. Kienzle (1988) [46] found that sucrase activity was higher in adult dogs than puppies if the

diet contained soy, lactose, and sucrose. However, the sucrase activity was similar between puppies and adult dogs if they were fed carbohydrate-free diets [46]. Therefore, sucrose may escape digestion and be fermented in the large intestine as a fermentable carbohydrate. Thus, the treatments with OS, WSBRSV, and WSBOS represented canine diets containing WSB depending on the dogs' sucrase activity levels in their small intestine. The WSBRSV and WSBOS resulted in more butyrate production than the BP, indicating that feeding WSB might have a beneficial impact on colonic health in dogs.

Isobutyrate and isovalerate are produced from fermentation of amino acids rather than carbohydrates [16,47]. According to [48], the fermentation of undigested protein yielded ammonia, valerate, and branched-chain fatty acids (isobutyrate and isovalerate) in dog feces. Panasevich et al. (2015) [49] observed no changes in markers of protein fermentation such as fecal branched-chain fatty acids with increasing soluble corn fiber (higher total dietary fiber) supplementation. These branched-chain fatty acids were generated when energy was limited in the large intestine [35]. According to [50], the absence of carbohydrates and the presence of undigested protein available in the hindgut could favor increased proteolytic activity by a greater number of bacteria. Detweiler et al. (2019) [51] found that no fiber treatment had significantly greater branched-chain fatty acids in dog feces compared to the addition of fiber treatments. Middelbos et al. (2007) [35] suggested that the rapid fermentation of fructooligosaccharides in the proximal colon in dogs might have resulted in the limited energy environment in the distal colon, leading to an increased catabolism of amino acids. Propst et al. (2003) [52] reported higher ammonia, isovalerate, and total biogenic amines in dog feces when the dogs received dietary fructans.

In our study, WSBOS had the highest ($p < 0.05$) isobutyrate and isovalerate concentrations. The valerate concentration was the highest ($p < 0.05$) for both WSBOS and WSBRSV. Considering the treatments were inoculated with the same population of anerobic bacteria, the reason for the highest branched-chain fatty acids in WSBOS could be explained by the rapid fermentation of the OS. The butyrate concentration of WSBOS seemed to be reach the maximum at an 8 h timepoint, showing a more rapid fermentation rate than the other treatments. Middlebos et al. (2007) [35] reported that OS are highly fermentable compared with fiber and are rapidly consumed once they enter the colon. Especially, valerate concentrations increased between 8 and 12 h of fermentation more so than other SCFA. Specific bacteria such as *Megasphaera elsdenii* are known to produce valerate along with acetate, propionate, and butyrate in pig intestines [53]. The *Megasphaera elsdenii* was in the Beagle dogs' fecal microflora [54], and these bacteria might have more actively produced valerate in late timepoints in the current study.

The PF had low concentrations of butyrate, whereas SH had butyrate concentrations similar to WSB and BP. Legume hulls contain large quantities of xylan as hemicellulose polymers [39], which were identified as part of IDF and NSP. The variation in degradability of the NSP was very large due to the different degrees of cell wall lignification, particle size, and retention time in the gut [41]. For a good fermentation of NSP in the colon, an adequate amount of nitrogen is required to feed colonic bacteria. In vivo, adequate levels of nitrogen are generally provided by residual undigested protein escaping the small intestine, endogenous nitrogen in mucus and epithelial cells, and urea recycled into the gastrointestinal tract [55]. In our in vitro system, an adequate amount of nitrogen was provided by yeast extract in the media solution. The PF TDF residues contained the least amount of CP, which might explain the lowest branched chain fatty acids concentrations. Lignin and crystallinity of cellulose in PF might have contributed to limiting the rate and extent of the microbial fermentation. To increase the fermentability of legume hulls, heat pretreatment or fiber-degrading multi-enzyme supplementation has been used in pigs [2].

According to their chemical composition and fermentative end-product concentrations, WSB can potentially be used as prebiotic ingredients based on two assumptions. Firstly, WSB contained high amount of TDF and OS that were indigestible by mammalian digestive enzymes but were fermented in the colon by the microbiome. Secondly, the WSBOS treatment, which represents the biological situation of dogs fed WSB if small intestinal

digestion of sucrose is low, showed more than twice the butyrate concentrations of the BP. Butyrate is oxidized by the intestinal mucosa and serves as the preferred energy substrate of colonocytes [56,57]. Moreover, the fermentation of nondigestible carbohydrates can affect the host by stimulating the growth and activity of beneficial bacterial concentrations (i.e., *lactobacilli* and *bifidobacteria*) and decrease potentially harmful bacteria (i.e., *Escherichia coli* and *enterobacteria*) in the gut [39]. Microbiota changes were not analyzed in the current study, which is a potential future research opportunity. However, further animal feeding studies are needed to determine the appropriate dose of WSB in dogs that have minimal anti-nutritional effects and flatulence induced by OS.

## 5. Conclusions

This work demonstrated that WSB has the potential as a prebiotic, yielding more butyrate production than BP in a canine in vitro fermentation model due to both fiber and highly fermentable OS. Further animal feeding studies are needed to determine the appropriate dose of WSB in dogs with measurements of canine health and microbial populations in the gut. On the other hand, PF was poorly fermented, having a high portion of IDF. This ingredient could be included in weight control diets anticipating the larger effect of IDF than gut health in dogs.

**Author Contributions:** Conceptualization, C.G.A.; methodology, H.S.K. and C.G.A.; software, H.S.K.; validation, C.G.A.; formal analysis, H.S.K.; investigation, H.S.K. and C.G.A.; resources, C.G.A.; data curation, H.S.K.; writing—original draft preparation, H.S.K.; writing—review and editing, C.G.A. and E.C.T.; visualization, H.S.K. and C.G.A.; supervision, C.G.A.; project administration, C.G.A. and H.S.K.; funding acquisition, C.G.A. All authors have read and agreed to the published version of the manuscript.

**Funding:** This research was funded by Kansas Soybean Commissions (sponsor award #2197).

**Institutional Review Board Statement:** The dog feeding experiment was approved by the Kansas State University Institutional Animal Care and Use Committee (IACUC) under protocol #4566.

**Informed Consent Statement:** Not applicable.

**Data Availability Statement:** Not applicable.

**Acknowledgments:** The authors thank MKC (Manhattan, KS, USA) for the donation of whole soybeans. The authors thank Prairie AquaTech (Brookings, SD, USA) for the donation of soybean oligosaccharides.

**Conflicts of Interest:** The authors declare no conflict of interest. The funders had no role in the design of the study; in the collection, analyses, or interpretation of data; in the writing of the manuscript; or in the decision to publish the results.

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
