# Peer review of "Evaluation of Fermentability of Whole Soybeans and Soybean Oligosaccharides by a Canine In Vitro Fermentation Model"

_fermentation, doi:10.3390/fermentation9050414_

Round 1
Reviewer 1 Report
Dear authors:
General comments
The presented paper is a very interesting document on the use of an in vitro fermentation model to examine the effect of faecal microbiota of dogs on the digestibility of different soybean compounds. However, I believe that the analysis of the microbiota changes in the in vitro fermentation tank, using deep sequencing methods, would have enriched the results section and could have helped to explain the results of the observed chemical changes, especially from SCFA.
Minor changes
Line 41: “Bifidobacterium” must be in italic type letter.
Line 51: Please, change “microflora” by “microbiota”.
Line 433: “E. coli” must be in italic type letter and in complete form “Escherichia coli”.
Author Response
Dear reviewer,
General comments:
The presented paper is a very interesting document on the use of an in vitro fermentation model to examine the effect of faecal microbiota of dogs on the digestibility of different soybean compounds. However, I believe that the analysis of the microbiota changes in the in vitro fermentation tank, using deep sequencing methods, would have enriched the results section and could have helped to explain the results of the observed chemical changes, especially from SCFA.
: Thank you very much for your comments and suggestions with regard to this manuscript. Yes, it would have been more valuable if we had analysis of the microbiota. We talked about it as a potential future research opportunity in Line 446-447. We will consider adding this parameter to our in vitro fermentation trials in the future.
Minor changes:
Line 41: “Bifidobacterium” must be in italic type letter.
Line 40: “Bifidobacterium” has been changed into italic.
Line 51: Please, change “microflora” by “microbiota”.
Line 50: “microflora” has been changed into “microbiota”.
Line 433: “E. coli” must be in italic type letter and in complete form “Escherichia coli”.
Line 445: “E. coli” has been changed into “Escherichia coli”.
Reviewer 2 Report
The subject matter is interesting. However, there are several things to revise, some points in the material and methods sections and discussions need to be better defined, and the results completely revised. Comments can be found in the attached pdf file.

Author Response
Dear reviewer,
General comments:
The subject matter is interesting. However, there are several things to revise, some points in the material and methods sections and discussions need to be better defined, and the results completely revised. Comments can be found in the attached pdf file.
: Thank you very much for your comments and suggestions with regard to this manuscript. We have addressed your comments in the revised manuscript.
Comments on pdf file:
Abstract: Please reported more results, too much space for introduction and material and methods. Delate the sentence about statistical analysis.
Line7-19: Thank you for the suggestions. Introduction and material methods became shorter, and more results have been added into the abstract.
Line 70: Please reported a reference for the protocol used.
Line 70-71: The reference (Donadelli et al., 2019) for the preparation procedures of fiber sources was added.
Line 77: Why this difference in granulometry compared to SH and PF?
Line 77: Very good point. The whole soybeans were ground by hammermill using a #16 standard sieve (1.191 mm) which is a common particle size feed. The SH and PF were included as controls and were from a previous study and had been received in the smaller particle size (0.5 mm). While ideally, we would have started with a common particle size amongst all treatments, this was not an option in the current study. Any differences in the results that might be associated with the particle size will be noted in the discussions and (or) limitations of the study.
Line 126: How long?
Line 127: The dogs were fed the commercial diet for 3 days during transition from one experiment to another. The previous experiment used a similar dietary framework in with the dogs had been fed for 70 days.
Line 127: Please provide a reference
Line 128: This has been changed to “to maintain the dogs body weight.”
Line 167: It is not little?
Line 173: The amount of sample (0.3g) was used for the in vitro fermentation following the published procedure described by (Donadelli et al., 2019).
Line 169: Normally the blank is tube with only inoculum.
Line 175-176: The blanks did not contain any fiber sources. The sentence has been revised for clarification.
Line 229: why?
Line 237: The blank (without any fiber sources) was included as a treatment to be used as a negative control and to correct for any artifact from inoculum in the experiment.
Line 239 (Table 1): This table refer to M&M or results, because you did not report any statistical differences in table so you can not discuss these data as results
Line 247: The statement has been deleted.
Line 247 (Table 4): Statistical analysis was not performed on these data, you can not discuss them
Line 255: Table 4 shows the analyzed values of the nutritional composition of fiber sources. The values are the average of the duplicates. The discussions regarding the nutritional compositions in Line 307-308 addressed the similarity of the analyzed values compared to the references. The word ‘numerically’ has been added in the Line 314 to clarify that the comparison was not derived based on the statistical analysis.
Line 255: It's a comment not a results
Line 264: The statement has been deleted.
Line 276: I don't understand, in table there are also 4-8-12 h
Line 285: The result statements were made based on the 12 h time point. The Table 6 shows all 4,8, and 12 h results to provide information for readers.
Line302-304: is not relevant in these sections.
Line 312: The statement has been moved to Line 343-346.
Line328 (4.3 short-chain fatty acids): too much references refered to pig trials.
Line 335: Thank you for the comment. As we checked the citations in 4.3 discussions, there were 13 dog trials, 8 human studies, 4 review articles, 4 pig trials, 2 rat studies, 1 domestic cat study, 1 chicken study, and 1 book reference. We understand your concerns that we did not use more dog trials as a reference; however, the dog’s data are still limited compared to human or monogastric animal research. Although 13 dog trials were referred to in this paper, more dog studies are needed to contribute to our current understanding of hindgut fermentation in dogs.
Round 2
Reviewer 2 Report
some doubts remain...
I still don't understand why you have included the blanks in the data analysis, such a choice may undermine the whole data analysis.
In addition, tab 4 is still within the results, these data have not been statistically analysed, so it cannot remain in that section, at most it must be reported within materials and methods.
Author Response
Comments and suggestions for authors:
Some doubts remain...
: Dear reviewer, thank you very much for your comments and suggestions with regard to this manuscript.
I still don't understand why you have included the blanks in the data analysis, such a choice may undermine the whole data analysis.
: The values from the blank tubes were needed for the calculation of both OMD and SCFA production rate. The use of the blanks for calculating OMD was described in the initial submission (Line 218-221). We have expanded our description of the use of the blank for the calculation of SCFA (Line 249-250). We’re glad the reviewer identified this issue so we could provide a better description of that calculation.
In addition, tab 4 is still within the results, these data have not been statistically analysed, so it cannot remain in that section, at most it must be reported within materials and methods.
: Text describing the composition of the substrates was moved to section 2.3 (Line 202-209).